# Forecasting the Future Excellence: 30 Years of Evaluating Service Organizations in Slovakia

Kristina Zgodavova [1,*], Peter Bober [2], Nataša Urbančíková [3], Gilberto Santos [4] and Andrea Sütőová [1]

1 Institute of Materials and Quality Engineering, Faculty of Materials, Metallurgy and Recycling, Technical University of Košice, Letná 9, 04200 Košice, Slovakia; andrea.sutoova@tuke.sk
2 Faculty of Electrical Engineering and Informatics, Technical University of Košice, Letná 9, 04200 Košice, Slovakia; peter.bober@tuke.sk
3 Faculty of Economics, Technical University of Košice, Letná 9, 04200 Košice, Slovakia; natasa.urbancikova@tuke.sk
4 Design School of Polytechnic Institute of Cávado and Ave (IPCA), 4750-810 Barcelos, Portugal; gsantos@ipca.pt
* Correspondence: kristina.zgodavova@tuke.sk; Tel.: +421-903-750-590

**Abstract:** The aim of this paper is to model and interpret the results obtained from the assessment of the Level of Excellence of Slovak service organizations using the criteria of the European Foundation for Quality Management (EFQM) excellence model. The Gompertz logistic function is effectively employed to fit the incremental improvement and predict the values of future Levels of Excellence. The EFQM model is usually used to improve organizational development and performance. The study focuses on the problem of the slow growth or even stagnation of Slovak service organizations towards Excellence. The questionnaire method was used to assess the Level of Excellence of the selected organizations, and the approach of measuring efficiency as a ratio of results and enablers was used to evaluate the organization's ability to transform inputs into outputs. Data were collected from 30 service organizations over a period of 20 years. The first finding of the study is the demonstration of the applicability of the Gompertz function to model the evolution of the Level of Excellence. The accuracy of the model is very high, and this predisposes this function to be used to forecast the scores of organizations over time. Examining efficiency yielded a second finding, that organizations were failing to capitalize on the effort put into translating it into results. After the first few years of growth, efficiency stagnates and then even declines. This suggests that the application of the original EFQM excellence model has reached the end of its ability to improve the effectiveness of organizations as a whole. Individual firms may have been growing or declining, but the average service score across the country had no longer the capacity to improve anymore.

**Keywords:** EFQM model; Gompertz function; forecasting; Level of Excellence; service organization

## 1. Introduction

Services are playing an increasingly important role in the national economy. They are already often the largest source of employment and job creation [1]. Transport services, tourism, energy services, and knowledge and information services have gradually replaced declining employment in industry and agriculture in developed countries. Services such as electronic banking, telecommunications companies' voice services, and postal services, which were previously considered "domestic", have become internationally tradable. Services account for more than two-thirds of the world's gross domestic product (GDP) and are the fastest growing sector [2]. The share of services in GDP is estimated to be 65% globally, with the EU-28 accounting for 66.7% in 2020. Western European countries in particular have a higher share [2,3].

Slovakia is a country with a relatively high share of industrial employment in the European Union, mainly due to the production of vehicles. However, the service sector

accounts for more than 60 percent of Slovakia's total GDP [4,5]. Services in Slovakia are dominated by trade and real estate, and dynamic development is in tourism, weakened by the COVID-19 pandemic. Public services, professional and support services, and ICT (information and communication technology) services are the next in terms of importance and size. As the country's industry is driven by the automotive and electronics sectors, domestically owned advanced exportable services deserve considerable policy attention as they could significantly help the economy grow.

Hence, service organizations are challenged to develop, improve, and innovate in the face of the global competitive environment [6]. One tool that can help them is the European Foundation for Quality Management (EFQM) Excellence model that employs the principle of self-assessment. This model is the most widely used tool for improving organizations in Europe and according to [7] is used by more than 50,000 organizations regardless of size or type [8]. Excellence is considered the highest level of quality and is the result of quality development efforts [9].

The score obtained from the self-assessment shows the level of maturity of the organization in various areas and allows for predicting the organization's performance [10,11]. Some literature sources have tested the possibility of modeling the evolution of scores over time using the Gompertz function [12,13]. In this context, this study aims to answer the following questions:

1.  Could the Gompertz function be used to predict the Level of Excellence of service organizations?
2.  To what extent are organizations able to transform enablers (what the organization does) into results (what the organization achieves) and improve the Level of Excellence?

## 2. Theoretical Framework

### 2.1. EFQM Excellence Model

The EFQM foundation was founded in 1989, and the first version of the EFQM Excellence model was created by a group of experts from various sectors and academic institutions and launched in 1992 [14]. This model provides the basis for awarding the European Quality Award (EQA) by the EFQM foundation to organizations that are the best examples of the Total Quality Management (TQM) in Europe.

The EFQM Excellence model can be used to identify how the various activities of an organization need to be coordinated to achieve the desired result. The basic principle of the model is self-assessment which consists of comprehensive, systematic, and regular reviews of the organization's activities and results that support the organization to achieve its desired goals [15]. The EFQM suggests a number of approaches for implementing the EFQM excellence model self-assessment (i.e., questionnaire, matrix chart, workshop, and pro-forma and award simulation) [16]. A self-assessment score (0–1000) ranks organizations in five levels of maturity [17]: starting (0–150)—early stage of developing and effectively run organization; progress (151–300)—some good practices but not an organization-wide cohesive approach and results moderate or not known in some areas; mature (301–450)—good results in most areas but innovation/industry leadership is lacking, strategies, systems, and people are not fully benchmarked and/or systematically improved; advanced (451–700)—an industry leader with strong results in comparison to benchmarks, and effective strategies, systems, and people in most areas; and world-class (701–1000)—supported by highly effective strategies, systems, and people.

The EFQM Excellence model is driven by the cause-and-effect relationships between the enablers (what the organization does) and results (what the organization achieves) [17]. The enablers consist of the following criteria: 1. leadership (maximum score 100), 2. strategy (maximum score 80), 3. people (maximum score 90), 4. partnership and resources (maximum score 90), and 5. processes (maximum score 140). The results are: 6. customer results (maximum score 200), 7. people results (maximum score 90), 8. society results

(maximum score 60), and 9. key performance results (maximum score 150). The maximum score of the criteria represents its weight in the assessment.

The underlying principle of the model is adding value for customers. The model also includes the RADAR dynamic assessment framework that takes into account the aspects of results, approach, deployment, and assessment and review. Thus, RADAR provides a multidimensional structured approach to validating organizational performance. Following the RADAR rationale, an organization needs to define at the highest level the outcomes it intends to achieve as part of its strategy. Accordingly, it is clear that managers should reflect on what aims they want to reach within the organization, at both strategic and tactical levels. The setting of objectives should enable clear and focused measures capable of achieving the expected results in the organization when they are implemented.

This simple but sophisticated and important logic provides a framework for effective planning—strategic and tactical—as well as for evaluating organizational performance [18,19].

The model was successively revised in 2000, 2010, and 2020. Although revisions were made, the overall structure of the model and the purpose of use remained the same until 2019. The first revision mainly brought a change in that organizations could choose the percentage weightings of the criteria according to their strategy. The revision in 2010 partly changed the naming of Criteria 5, processes, products, and service; and 9, business results; and also clarified the content of the sub-criteria. In addition, "Fundamental Concepts of Excellence" was amended, containing eight principles: adding value for customers; creating a sustainable future; developing organizational capability; harnessing creativity and innovation; leading with vision; inspiration and integrity; managing with agility; succeeding through the talent of people; and sustaining outstanding results [20].

The latest revision of the EFQM model is valid from 2020. The new model, created using the "Design Thinking" methodology, has changed from a simple self-assessment tool to a tool that also includes a transformation methodology [21]. A comparison of the EFQM 2020 model with the previous version of 2013 can be found in [22–24]. The impact of the EFQM model on quality development in Europe has been very significant in recent years and has become the standard model for many national quality award schemes in European countries and has reached beyond Europe to the Middle East, Asia, South America, and South Africa [7]. Top performing European organizations also score above 700 in services [18,25].

Potential limitations in the use of excellence models, according to [16], may be a lack of understanding of the meaning and potential benefits of self-assessment for the organization, overly complex evaluation criteria, excessive paperwork, cumbersome procedures, and inadequate infrastructure.

*2.2. Questionnaire Method*

The EFQM Excellence model suggests five possible self-assessment methods: questionnaire, matrix, workshop, pro-forma, and award simulation. Experts agree that the decision about which self-assessment technique to choose depends on the culture of the organization, complexity of operations, levels of staffing, etc. [26]. As self-assessment was not a standard part of the processes in service organizations at the beginning of our research, a straightforward questionnaire method was chosen. The questionnaire survey is aimed at obtaining the opinions of managers and other employees of the organization about the Level of Excellence according to the respective criteria of the EFQM model. Training of both management and staff is required prior to the use of the questionnaire [7]. The advantages of this approach are its relatively fast and easy application, its possibility to involve all people in the organization, its support for communication efforts, but also the possibility to use it in conjunction with other self-assessment methods. A related risk is that respondents' views on strengths and areas for improvement cannot be directly ascertained. The accuracy of the self-assessment depends on the understanding of the questions, may be influenced by the subjective opinion of the respondents or even by efforts to manipulate the result, and fatigue may occur when completing the questionnaire. Support from the organizers is

essential to obtain valid responses [27]. The results of the application of the EFQM model of self-assessment using a questionnaire approach in services are presented, for example, in [28–33].

### *2.3. Gompertz Function*

Growth S-curve models with the Gompertz function have been used to predict performance in various fields of applied research, and originally Gompertz used it in the biological field to specify mortality law and life span [34]. In marketing, the Gompertz function has been used to model new product sales and predict market growth [35–37]. Other applications have been in management [38] and economics [39]. A group of S-curves is primarily used to visualize the progress of a measured parameter over time, which resembles the shape of the letter S. Growth is rapid at the beginning but gradually slows towards its perceived limit of improvement. The logistic curve is used to show a symmetrical progression. However, in the case of measuring excellence, there is an obvious asymmetry of development, which corresponds to the employment of the Gompertz curve [40].

Despite their limitations, S-curves have also been extensively used in technology management to model technology performance limits [12]. Mlčoch and Slimák used the Gompertz function for the evaluation of the bearings' quality [41], and Zgodavová & Slimák used it for the evaluation of the production quality in general [13]. The Gompertz model also appeared in the work of [12] as a benchmarking and self-assessment tool and a piecewise regression model for sustainable business excellence. In a study [11], the possibility of applying the Gompertz model to predict the growth of integrated management system implementation in organizations is presented.

Gompertz's model was formulated as a solution to a differential equation that assumes that the population growth rate is a function of the logarithm of the saturation limit [42]. Although the Gompertz curve is similar to the simple logistic curve, it is not symmetric concerning its inflection point. The inflection point for the Gompertz model is approximately at 37% of the long-term saturation levels [43]. Several different re-parameterizations of the traditional cumulative Gompertz model exist [37,44]. One of them is given by Equation (1).

$$Y(t) = ae^{-e^{(-k(t-T_i))}} \tag{1}$$

where $t$ denotes time, $a$ is an asymptote (saturation level), $e$ is Euler's number, $k$ is a growth-rate coefficient, and $T_i$ represents a time at an inflection point where the speed of growth is maximal. The $T_i$ parameter shifts the growth curve horizontally without changing its shape (allocation parameter), whereas $a$ and $k$ are shape parameters that affect curve shape [44,45].

One special parameter $b$ was added to Equation (1) that represents the entry level of the s-shaped evolution step. This parameter shifts the curve vertically. The final saturation level is the sum of $b$ and original Gompertz curve saturation $a$. This modification reflects the fact that the entry level of the EFQM Excellence model score is higher than zero [46]. Hence, the modified equation is as follows (2):

$$Y(t) = b + ae^{-e^{(-k(t-T_i))}} \tag{2}$$

## 3. Methodology

### *3.1. Context of the Study*

The starting point of the research was the EFQM Excellence Model and the history of the National Quality Award of the Slovak Republic (NQA SR). One of the preconditions for the accession of the Slovak Republic to the European Union was also the implementation of the document adopted by the Council of EU ministers about the "European Quality Promotion Policy" of 1994. The European Commission required the drafting of National Quality Programmes by each member state in the Union. The government of the Slovak Republic, as one of the few associated countries, passed Resolution No. 673/1998, declaring the National Quality Programme of the Slovak Republic (NQA SR) until 2003. This program

also included the declaration of 2000 to be the Year of Quality and the implementation of NQA SR as one of the top priorities of the program [28]. The government of the Slovak Republic continues to pursue this program and passed further resolutions related to the NQA SR in 2004–2008, 2009–2012, and 2013–2016. The National Quality Award of the Slovak Republic is the most prestigious quality award for the Slovak organizations. Thus, NQA is the highest award to be gained nationally, ultimately enabling the winner to gain national recognition amongst competitors in terms of quality management [4].

Slovakia officially launched awarding prizes in several categories, including services, in 2000. In the same year, the data collection for this study also started; therefore, the whole evolution of the EFQM application is available from the very beginning. Due to the COVID-19 pandemic situation, the NQA SR was not held in 2019 and 2020. Our research team applied the EFQM model to assess the Level of Excellence in 30 selected Slovak service organizations, which we tracked over 20 years using the Self-Assessment for Quality Improvement (SAQI) software tool we developed for data collection. The purpose of the software tool was to overcome some cumbersome procedures when collecting data.

### 3.2. Sample and Data Collection

The sample of organizations is representative of the types of services, and contacts were made at themed exhibitions where our team worked to evaluate and award the best service providers. The survey population consisted of 30 service organizations that had undergone self-assessment over a 20-year period following the EFQM Excellence criteria. The survey included service providers in the energy, food service, hospitality, information and communications technology, healthcare, and education sectors of varying sizes. Some of the surveyed organizations participated in the competition for the National Quality Award of the Slovak Republic.

Firms were motivated to be involved in the longitudinal quality assessment because of the opportunity to benchmark themselves and to win a national quality award in the service category. If a firm was dissolved or lost interest, it was replaced by a firm with a comparable profile by industry and size. Accordingly, the results of the self-assessment are reliable and valid sources of information due to the training, specialization, and qualifications of the interviewers involved in the process.

The data were collected using specialized software that allowed longitudinal historical data collection, analysis, and evaluation of the obtained data. The software also included educational material used to provide training to those conducting the self-assessment. We first tested the questionnaire on a pilot sample of 10 respondents and, after modification, used it for self-assessment in organizations. After the update of the EFQM Excellence Model in 2010, we refined the questions on Criteria 5 and 9 in the questionnaire. Subsequently, we tested it again to maintain the continuity of the assessment. From 2000 to 2019, 600 assessment reports with complete and valid results were collected.

### 3.3. Measures

The score was measured using indicators from 32 sub-criteria grouped to 9 criteria of the EFQM Excellence Model (Appendix A, Table A1). The measurement scales of the RADAR scoring matrices were used for all sub-criteria to obtain the scores for each indicator. Appendix B, Tables A2 and A3 show the RADAR matrix for enablers and for results, respectively. The score for each criterion is the product of the average of the sub-criteria scores and the criterion weight. The total score represents the sum of enablers and results (Appendix C, Table A4).

### 3.4. Data Analysis

This study applied the graphical presentation of the data and the Gompertz function to interpret the results and predict the trend toward excellence. The Gompertz function parameters were calculated by the trust-region algorithm for curve fitting, and the model precision is assessed by the calculation of the adjusted coefficient of determination [37].

The ability to convert enablers into results was investigated using the efficiency indicator (EFF) [47], which is defined by the Formula (3).

$$\text{EFF} = \frac{\text{Value of Output (Results)}}{\text{Value of Input (Enablers)}} \tag{3}$$

## 4. Results and Discussion

Figure 1 shows the average score of the 30 service organizations assessed according to the EFQM Excellence model criteria for the three time periods with the fitted Gompertz function.

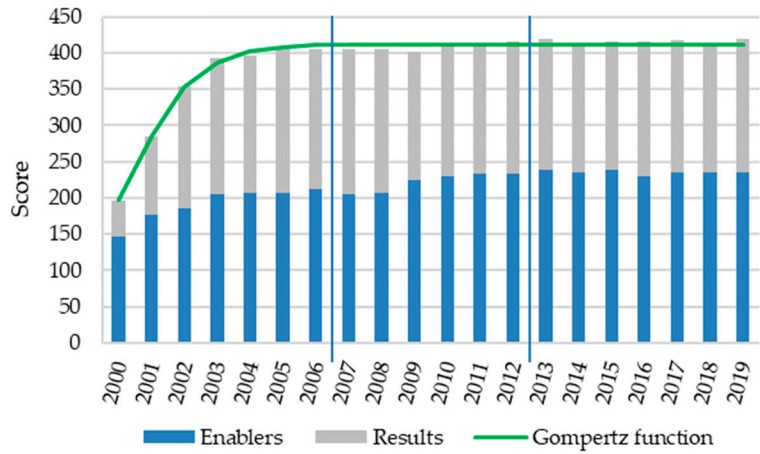

**Figure 1.** Average score of 30 service organizations and the corresponding Gompertz function ($b$ = 142.04, $a$ = 270.18, $k$ = 0.9281 $T_i$ = 2000.51) during three time periods.

The years 2000–2006 saw a sharp improvement in the score; 2007–2012 was a period of slight deterioration after the onset of the global financial crisis, followed by a period of new growth due to the recovery. The years 2013–2019 were a period of stagnation. In the first period, both enablers and results were still growing. The next period, marked by the economic crisis of 2008–2009, saw enablers growing but results declining. In the last period, both enablers and results stagnated. The model's ability to stimulate growth has been exhausted.

### 4.1. Using Gompertz function for the Service Organizations Level of Excellence Forecasting

The Gompertz function proved to be a suitable model to describe the existing evolution of the Level of Excellence according to the EFQM Excellence model; the accuracy of the model is very high. The value of the adjusted coefficient of determination $R^2_{adj}$ is 0.9907 (recalculated from initial value $b$ = 142.04) proves the high explanatory power of the Gompertz trend. To validate the suitability of this model for forecasting, we use data from the early years retrospectively to build the model and check the forecast against reality in subsequent years (Figure 2).

Figure 2 illustrates the ability of the Gompertz function to predict the future Level of Excellence. Data from three years 2000–2002 were sufficient enough to reliably predict the saturation four years later. However, we assume that the achievement of high forecasting accuracy was reached due to the circumstances:

- The first three years covered the informative period before and after the maximum growth (inflection point of the Gompertz curve).
- The average of 30 companies filtered out random deviations and exploited the validity of "Central Limit Theorem" known from statistics.
- No extreme events occur in the next four years that would invalidate the forecast.

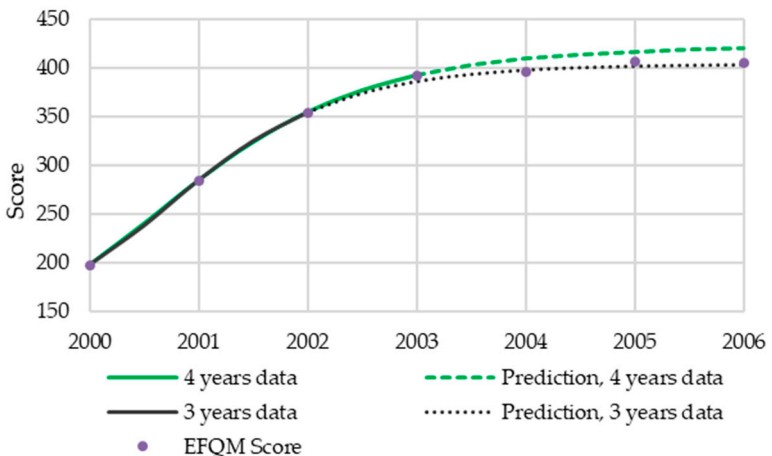

**Figure 2.** Level of Excellence forecasting using data from three and four years.

Estimation of the four unknown parameters of the Gompertz function requires four independent equations: the input of four data points. In Figure 2, we use three data points and the entry score level $b$ was set to a value of 160 by an educated guess from prior knowledge. The deviation of the predicted value from the actual one was 1.51% in 2006. The variation of $b$ in a range between 140 and 190 changes the deviation from 4.90% to −5.97%. Using four data points from 2000 to 2003 leads to a deviation of 3.7%. This shows that the forecasting model can tolerate imprecision of an educated guess, and three years of measurement should be enough to forecast the score saturation in four years. The overview of four Gompertz functions parameters and score deviations are in Table 1.

**Table 1.** Parameters of Gompertz functions and score deviations.

| Description | Data Points | $b$ Calculated $b$ Preset Value | $a$ | $k$ | $T_i$ | Score in 2006 | Deviation in 2006 |
|---|---|---|---|---|---|---|---|
| Measured score in 2006 | | | | | | 405.00 | - |
| Three years, $b = 160$ (Figure 2) | 3 | **160.00** | 252.35 | 0.9940 | 2000.66 | 411.11 | 1.51% |
| Three years, $b = 140$ | 3 | **140.00** | 287.66 | 0.8499 | 2000.57 | 424.83 | 4.90% |
| Three years, $b = 190$ | 3 | **190.00** | 190.88 | 1.5405 | 2000.78 | 380.82 | −5.97% |
| Four years (Figure 2) | 4 | 147.54 | 274.65 | 0.8965 | 2000.60 | 420.02 | 3.71% |

### 4.2. The Ability of Service Organizations to Transform Enablers into Outputs

A detailed view of the flow and internal structure of enablers and results is shown in the Figure 3a,b. The Figure shows the average level of individual enablers sub-criteria (a) and results sub-criteria (b).

In the first onset period, all criteria in enablers grew because the EFQM approach represented a new perspective on the functioning of emerging service companies and had the ability to discover large gaps in strategy, leadership, processes, etc. In the second period, firms were under pressure from the global crisis, which was most evident in the short-term decline in Criterion 3 (people), but then it rose back by leaps and bounds after the crisis. Criteria 1 (leadership), 2 (strategy) and 4 (sources and partnership) continued to grow but only slowly. Criterion 5 (processes, product and services) had previously differed from the others by substantially higher values, but in the second period, they were the most affected by the crisis, and its performance was declining for the longer period. In the third period, after recovering from the crisis, the relationships between the criteria bounced back. Only Criterion 5 (processes, product and services) increased and reached the pre-crisis level and was able to slowly grow. Again, it deviated higher from the other criteria related to strategy, leadership, or people. However, all the other four criteria hit their natural limits and arrived at stagnation or even weakened, with a lack of capacity for future improvement.

Results of Criterion 6 (customer results) gradually grew during the first period, but in the second period, there was more or less stagnation, and the criterion declined in the third period. Criterion 7 (people results) grew sharply in the first period, declined sharply in the second period after the economic crisis, and started to grow slowly in the third period. Criterion 8 (society results) gradually grew, declined slightly in the second period, and grew very slowly in the third period. Finally, Criterion 9 (business results) grew sharply and reached saturation in the first phase, declined slightly after the crisis, and then cycled up and down.

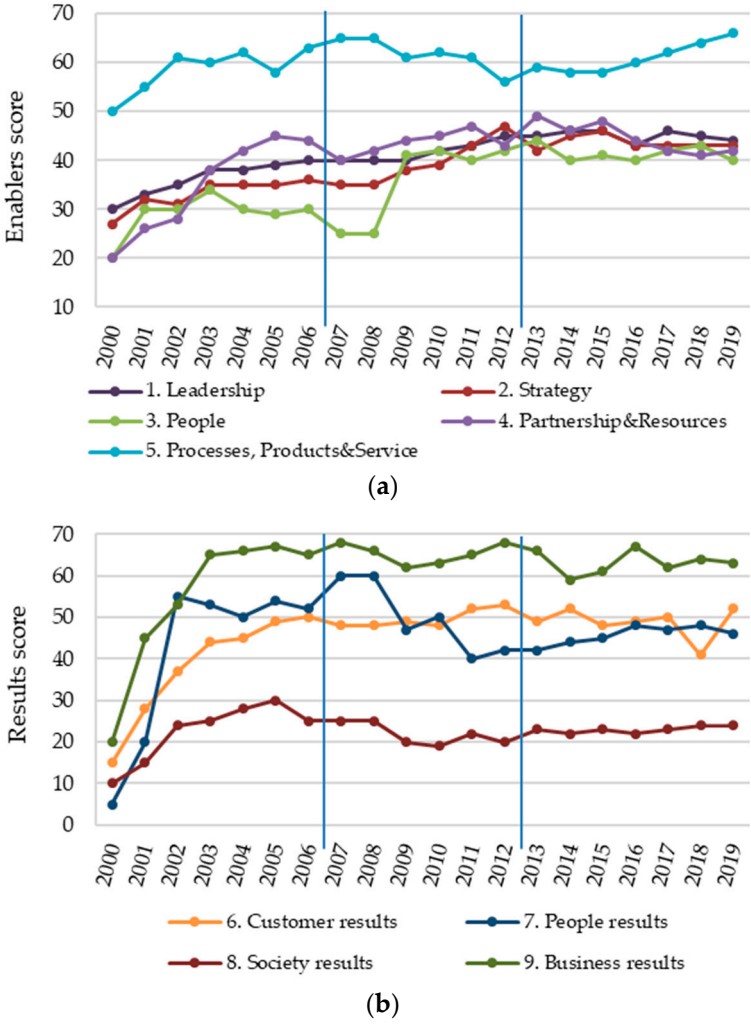

**Figure 3.** Decomposed Level of Excellence for: (**a**) enablers criteria; (**b**) results criteria.

The relationship between results and enablers is presented as the efficiency according to Equation (3) in Figure 4a, and as a scatterplot in Figure 4b for the three time periods. Figure 4a additionally contains the historical evolution of real GDP per capita [48] and the employment rate [49].

Efficiency grew sharply at the beginning of the first period showing positive effect of the EFQM but then began to stagnate. In the second period, the crisis manifested itself in a decline, and in the third period, efficiency became stagnant again. The scatterplot in Figure 4b reflects the correlation between the values of results and enablers. In the first period, there is a positive correlation, when enablers increase, so do results. The second and third periods show a negative correlation, i.e., despite increasing enablers, results are decreasing. Finally, in the third period, results fall more sharply as enablers rise than in the second period, as is evident from the coefficients of the regression lines in Figure 4b.

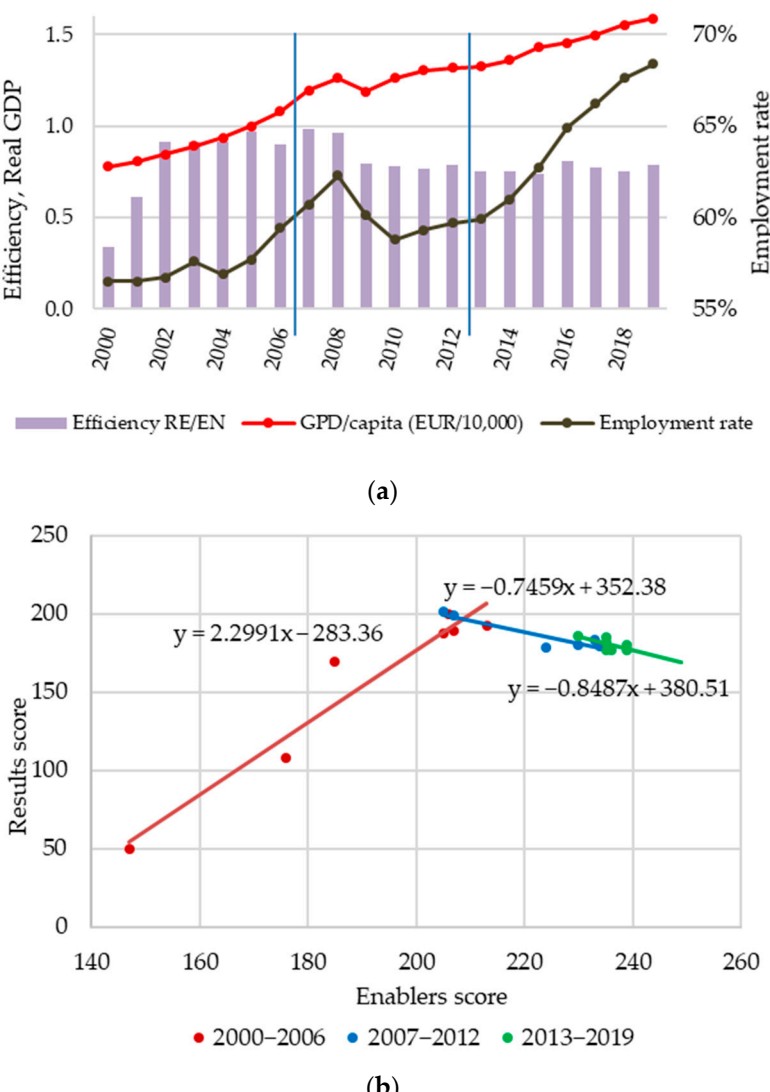

**Figure 4.** Relationship between results and enablers: (**a**) efficiency, real GDP per capita and employment; (**b**) scatterplot of results against enablers.

The period up to 2000 is characterized by a turbulent transition from a centrally controlled national economy to a market economy. The year 2000 was declared the Year of Quality in Slovakia, and the awarding of the National Quality Award [4] was launched in line with the EFQM model of excellence. A large group of companies, after media coverage and extensive training, started to use this tool for their improvement. In addition, the Service Quality Award was first awarded in 2000. Hence, there is a sharp growth in efficiency in 2001–2002 against the backdrop of economic recovery and rising real GDP per capita in the country. Service organizations were able to learn and make intensive use of their enablers to promote results. This fact is confirmed by the positive correlation evident in Figure 4b. However, in the following years, the growth of efficiency stalled despite the continued growth of GDP. We conclude that the potential of the EFQM tool has been depleted under the given conditions.

Although in the second reporting period the overall EFQM score remained approximately at the same level, efficiency dropped significantly. The economic crisis in 2008–2009 meant that the ability to convert enablers into results declined and has not bounced back to the 2007 level. GDP has also fallen in line with this, and its growth has stalled. Two years after the crisis, the employment rate also slumped and only started to grow in 2014. Between 2007 and 2012, a number of foreign companies came to Slovakia, bringing a new

corporate culture to service organizations as well. In the meantime, many organizations have also introduced an ISO 9001 quality management system and Six Sigma methodology [50,51], and some an ISO 14001:2015 environmental management system. Several organizations have started to apply for the Quality Award for Services [4]. This situation was also reflected in the growth of the scores: 1. leadership, 2. strategy, 3. people, and 4. partnership and resources according to Figure 3a. However, the increase in enablers and especially investing in 3. people, was not reflected at all in results. On the contrary, Criteria 7. people results, with the sub-criteria perception measure and performance and 8. society results, declined. The crisis brought a loss of perspective and motivation for personal improvement and social security (employment declined). The consequence, especially in the field of health and social services, has been a move abroad, and the costs spent on education have not been recouped.

The situation in efficiency did not change significantly in the third period either; the correlation between enablers and results was negative in both periods as shown in Figure 4b. In 2015, there were some clear signs of accelerated economic growth and improvement in the parameters of macro-economic stability or socio-economic parameters. Although both GDP and employment grew and overall social conditions improved, the ability to convert enablers into results has been lost. One reason for this could be that catching up with the most advanced economies represented by the former EU-15 has slowed down [36]. Another reason is that economic migration continued during this period.

The analysis and discussion show that the EFQM excellence model has definitely exhausted its ability to improve the efficiency of the organization. Individual firms may were growing or declining, but the average score in services in Slovakia was not improving and did not reach the total score comparable to other European countries, e.g., Germany. There, healthcare organization efficiency was 0.81 [52], while efficiency in Spain was about 0.71 [53,54]. The observed firms tried to leverage enablers but lacked the ability to drive outcomes, thus demonstrating a low ability to innovate and be competitive.

For comparison, we also present the EFF indicators of the EFQM Global Excellence Award (GEEA) applicants and the National Quality Award of the Slovak Republic (NQA SR) applicants for 2016 and 2019. We calculated efficiency according to Formula (3) as the average of all categories. GEEA applicants score at the advanced level (451–700), and NQA SR applicants score at the mature level (301–450). In both cases, the award simulation approach was used in the evaluation.

Efficiency for GEEA according to [55] in 2016 was 0.88 and in 2017 was 0.75, while the average total score in 2016 was 653 and in 2017 was 692. Efficiency for NQA SR candidates in 2016 was 0.81 and in 2017 was 0.73. The average total score in 2016 was 417 and in 2017 was 377.

## 5. Conclusions

The EFQM model enables self-assessment across multiple criteria and provides an opportunity to become aware of the functioning of service organizations in terms of effectiveness. The survey carried out in Slovakia suggests that there is a lack of sufficient systemic insight across their organizational levels and an understanding of mechanisms and linkages. Moreover, the EFQM model itself is being transformed into a new systems approach that facilitates a better awareness of inter-relationships within the system and with its environment. It remains an open question for further study as to how organizations will be able to adopt the systems approach in self-evaluation and use it to the benefit of enhancing excellence.

The results of the study demonstrated two main findings:

- The Gompertz function has confirmed its ability to describe the evolution and predict the Level of Excellence scores in the service organization segment in Slovakia over 20 years.
- The firms' overall scores were close to their limit and could not move further up without creative destruction. However, the price at which firms were able to maintain their

overall score—through higher investment in enablers—is also important. However, even though the enablers' benchmarks were raised proactively, they did not result in improved scores but instead in a slight decline. The trend in the efficiency of service businesses has reversed, and the predicted results have ceased to be achieved.

Modelling with a Gompertz curve has been confirmed as appropriate; the value of the adjusted coefficient of determination $R^2$adj is 0.9907, which corresponds to the high accuracy of the model. The average of 30 companies had the ability to filter out random deviations and showed effects of the Central Limit Theorem for sufficient sample sizes. Therefore, the Gompertz function can be used for predicting scores of organizations over time. Even scores from three or four years are sufficient to predict for the next few years. The curve has a fast-rising slope in the first phase, which gradually slows down in growth. Therefore, the use of the Gompertz trend was particularly suitable in the situation of Slovakia, when the EFQM framework was only being introduced, and companies had large margins and thus the ability to grow rapidly in the criteria monitored.

The second finding relates to the efficiency of organizations that have failed to capitalize on the efforts made to translate it into results towards the end of the observed period. After the first years of growth, efficiency experiences stagnation and even subsequent decline. This implies that the EFQM Excellence model in use has lost its capacity to improve efficiency. This negative trend of efficiency can eventually be overcome by changing the evaluation model; therefore further research will continue based on the new FQM Excellence Model [14] which is challenging organizations to build a new culture and improve their performance. Slovak service companies achieved particularly low societal results, which indicate too strong a focus on the company. Indeed, the typical culture of companies is too results-oriented and underestimates societal outcomes, focusing on improving the whole of society. In this case, however, it is necessary to achieve a cultural change in the view of entrepreneurship, which is certainly not easy and requires a longer period of time to deploy a new framework.

Possible limitations of these study results may be affected by the evaluators' own perception of the Level of Excellence.

**Author Contributions:** Conceptualization, K.Z.; methodology, K.Z., P.B.; software, P.B.; validation, N.U., G.S.; formal analysis, N.U.; investigation, K.Z.; resources, N.U., G.S., A.S.; data curation, K.Z., N.U.; writing—original draft preparation, K.Z.; visualization, P.B.; project administration, K.Z.; funding acquisition, K.Z. All authors have read and agreed to the published version of the manuscript.

**Funding:** This research was funded by the MINISTERSTVO ŠKOLSTVA, VEDY, VÝSKUMU A ŠPORTU SLOVENSKEJ REPUBLIKY, Grant No. KEGA 018TUKE-4/2022 Creation of new study materials, including an interactive multimedia university textbook for computer-aided engineering activities and VEGA 1/0633/20 Research of the variability of properties and functions of products made of composite materials produced by additive manufacturing.

**Institutional Review Board Statement:** Not applicable.

**Informed Consent Statement:** Not applicable.

**Data Availability Statement:** Not applicable.

**Conflicts of Interest:** The authors declare no conflict of interest.

## Appendix A

**Table A1.** EFQM Excellence Model criteria and sub-criteria.

| | No | Criteria | Subcriteria |
|---|---|---|---|
| Enablers | 1 | Leadership | 1a How leaders develop the mission, vision, and values and are role models for a culture of excellence in the organization<br>1b How leaders are personally involved in ensuring the organization's management system is developed, implemented, and continuously improved<br>1c How leaders are involved with customers, partners, and representatives of society<br>1d How leaders motivate, support, and recognize the organization's people |
| | 2 | Strategy | 2a How policy and strategy are based on the present and future needs and expectations of stakeholders<br>2b How policy and strategy are based on information from performance measurement, research, learning, and creativity-related activities.<br>2c How policy and strategy are developed, reviewed, and updated<br>2d How policy and strategy are deployed through a framework of key processes<br>2e How policy and strategy are communicated and implemented |
| | 3 | People | 3a Human resources plan<br>3b People's capabilities<br>3c Empowerment<br>3d Communication<br>3e Reward and recognition |
| | 4 | Partnerships & Resources | 4a Partnerships<br>4b Technological support for processes<br>4c Sustainability<br>4d Technology<br>4e Knowledge sharing |
| | 5 | Process, Products & Services | 5a Management and improvement of key processes<br>5b Innovation<br>5c Marketing and promotion<br>5d Production/delivery/service<br>5e Relationship management |
| Results | 6 | Customer results | 6a Perception Measures<br>6b Performance Indicators |
| | 7 | People results | 7a Perception Measures<br>7b Performance Indicators |
| | 8 | Society results | 8a Perception Measures<br>8b Performance Indicators |
| | 9 | Business results | 9a Key Performance Outcomes<br>9b Key Performance Indicators |

## Appendix B

**Table A2.** RADAR scoring matrix for enablers.

| RADAR Element | Attribute | Measurement Scale | | | | |
|---|---|---|---|---|---|---|
| | | 0% | 25% | 50% | 75% | 100% |
| Approach | Sound:<br>Approach has a clear rationale<br>Approach has a defined processes<br>Approach focuses on stakeholder needs | ○<br>○<br>○ | ○<br>○<br>○ | ○<br>○<br>○ | ○<br>○<br>○ | ○<br>○<br>○ |
| | Integrated: Approach is linked with other approaches as appropriate; Approach supports policy and strategy. | ○ | ○ | ○ | ○ | ○ |

**Table A2.** *Cont.*

| RADAR Element | Attribute | Measurement Scale | | | | |
|---|---|---|---|---|---|---|
| | | 0% | 25% | 50% | 75% | 100% |
| Deployment | Implemented: Approach is implemented. | ○ | ○ | ○ | ○ | ○ |
| | Systematic: Approach is deployed in a structured way with the method used for deployment being planned and executed soundly. | ○ | ○ | ○ | ○ | ○ |
| Assessment and Review | Measurement: Regular measurement of the effectiveness of the deployment is carried out; Measures selected are appropriate. | ○ | ○ | ○ | ○ | ○ |
| | Learning is used to: Identify best practice and improvement opportunities. | ○ | ○ | ○ | ○ | ○ |
| | Improvement: Output from measurement and learning is analyzed and used to identify, prioritize, plan, and implement improvements. | ○ | ○ | ○ | ○ | ○ |

0%—no evidence or anecdotal; 25%—some evidence; 50%—Evidence; 75%—Clear evidence; 100%—Comprehensive evidence.

**Table A3.** RADAR scoring matrix for results.

| RADAR Element | Attribute | Measurement Scale | | | | |
|---|---|---|---|---|---|---|
| | | 0% | 25% | 50% | 75% | 100% |
| Result | Trends: Trends are positive and good performance continues. | ○ | ○ | ○ | ○ | ○ |
| | Objectives: Objectives are agreed and are reasonable. | ○ | ○ | ○ | ○ | ○ |
| | Benchmarking: Results are reported. | ○ | ○ | ○ | ○ | ○ |
| | Causes: Results are achieved through approaches. | ○ | ○ | ○ | ○ | ○ |
| | Scope: Results are achieved in relevant areas. | ○ | ○ | ○ | ○ | ○ |

## Appendix C

**Table A4.** Measured scores, calculated Efficiency, and Gompertz function values.

| Criterion | Enablers (EN) | | | | | | Results (RE) | | | | | Total Score | Efficiency | |
|---|---|---|---|---|---|---|---|---|---|---|---|---|---|---|
| | 1. | 2. | 3. | 4. | 5. | EN Sum | 6. | 7. | 8. | 9. | RE Sum | | EFF = RE/EN | Gompertz Function |
| Year | | | | | | | | Score | | | | | | |
| 2000 | 30 | 27 | 20 | 20 | 50 | 147 | 15 | 5 | 10 | 20 | 50 | 197 | 0.34 | 196.66 |
| 2001 | 33 | 32 | 30 | 26 | 55 | 176 | 28 | 20 | 15 | 45 | 108 | 284 | 0.61 | 285.66 |
| 2002 | 35 | 31 | 30 | 28 | 61 | 185 | 37 | 55 | 24 | 53 | 169 | 354 | 0.92 | 352.50 |
| 2003 | 38 | 35 | 34 | 38 | 60 | 205 | 44 | 53 | 25 | 65 | 187 | 392 | 0.91 | 386.82 |
| 2004 | 38 | 35 | 30 | 42 | 62 | 207 | 45 | 50 | 28 | 66 | 189 | 396 | 0.91 | 401.88 |
| 2005 | 39 | 35 | 29 | 45 | 58 | 206 | 49 | 54 | 30 | 67 | 200 | 406 | 0.97 | 408.09 |
| 2006 | 40 | 36 | 30 | 44 | 63 | 213 | 50 | 52 | 25 | 65 | 192 | 405 | 0.90 | 410.58 |
| 2007 | 40 | 35 | 25 | 40 | 65 | 205 | 48 | 60 | 25 | 68 | 201 | 406 | 0.98 | 411.57 |
| 2008 | 40 | 35 | 25 | 42 | 65 | 207 | 48 | 60 | 25 | 66 | 199 | 406 | 0.96 | 411.97 |
| 2009 | 40 | 38 | 41 | 44 | 61 | 224 | 49 | 47 | 20 | 62 | 178 | 402 | 0.80 | 412.12 |
| 2010 | 42 | 39 | 42 | 45 | 62 | 230 | 48 | 50 | 19 | 63 | 180 | 410 | 0.78 | 412.18 |
| 2011 | 43 | 43 | 40 | 47 | 61 | 234 | 52 | 40 | 22 | 65 | 179 | 413 | 0.76 | 412.21 |
| 2012 | 45 | 47 | 42 | 43 | 56 | 233 | 53 | 42 | 20 | 68 | 183 | 416 | 0.79 | 412.22 |
| 2013 | 45 | 42 | 44 | 49 | 59 | 239 | 49 | 42 | 23 | 66 | 180 | 419 | 0.75 | 412.22 |
| 2014 | 46 | 45 | 40 | 46 | 58 | 235 | 52 | 44 | 22 | 59 | 177 | 412 | 0.75 | 412.22 |
| 2015 | 46 | 46 | 41 | 48 | 58 | 239 | 48 | 45 | 23 | 61 | 177 | 416 | 0.74 | 412.22 |
| 2016 | 43 | 43 | 40 | 44 | 60 | 230 | 49 | 48 | 22 | 67 | 186 | 416 | 0.81 | 412.22 |
| 2017 | 46 | 43 | 42 | 42 | 62 | 235 | 50 | 47 | 23 | 62 | 182 | 417 | 0.77 | 412.22 |
| 2018 | 45 | 43 | 43 | 41 | 64 | 236 | 41 | 48 | 24 | 64 | 177 | 413 | 0.75 | 412.22 |
| 2019 | 44 | 43 | 40 | 42 | 66 | 235 | 52 | 46 | 24 | 63 | 185 | 420 | 0.79 | 412.22 |

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
