# Peer review of "Forecasting the Future Excellence: 30 Years of Evaluating Service Organizations in Slovakia"

_applsci, doi:10.3390/app12146856_

Round 1
Reviewer 1 Report
The article models and interprets the results obtained from the assessment of the Level of Excellence of Slovak service organizations using the criteria of the EFQM model. In particular, the Gompertz-logistic function is employed effectively fit the incremental improvement and predict the values of future Levels of Excellence. A questionnaire method was used to assess the Level of Excellence of the selected organizations and the approach of measuring efficiency as a ratio of Results and Enablers was used to evaluate the organization's ability to transform inputs into outputs. Data was collected from service organizations over a period of 20 years using SAQI software tool. The first finding of the study is the demonstration of the applicability of the Gompertz-function to model the evolution of the Level of Excellence. On examining the efficiency it is noticed that the organizations were failing to capitalize on the effort put into translating it into results. After the first few years of growth, efficiency stagnates and then even declines. This suggests that the application of the original EFQM excellence model has reached the end of its ability to improve the effectiveness of organizations as a whole.
Generally the article is interesting but the following points should be addressed.
Avoid abbreviations in the abstract.
The language of the article should be improved.
It is not clear why Gompertz function is focused in the study as there are many competitive models.
The presentation of figures must be improved.
Figure 4(b), isnt it better to fit nonlinear model than a liear regression?
Author Response
Dear reviewer 1,
Thank you for your valuable review. We have tried to take all your comments into account.
Reviewer 1: Avoid abbreviations in the abstract.
Response: abbreviations in the abstract are in full for better reading comprehension.
Reviewer 1 and 2: English language and style are fine/minor spell check required.
Response: the typos in the article have been corrected and the text has been proofread by a native English language expert.
Reviewer 1: It is not clear why Gompertz function is focused in the study as there are many competitive models.
Response: A separate section 2.3 is devoted to the Gompertz curve, justifying its use for modelling and forecasting. However, we have provided new justifications. Unlike the logistic curve, which is used to represent symmetrical growth, the Gompertz curve is asymmetric, which better corresponds to the measured data.
Reviewer 1: The presentation of figures must be improved.
Response: we have redesigned the figures to comply with the standards of the MDPI Applied Sciences journal.
Reviewer 1: Figure 4(b), isn't it better to fit nonlinear model than a liear regression?
Response: We understand the question, the progression under unchanged external conditions would resemble a non-linear curve. However, the essence of the figure is the shift caused by an external shock (the 2007 global financial crisis) that changed the regression line to a different one.
The Figure 4(b) is a scatterplot that shows the relationship between two numerical variables. Linear regression was used because it shows a correlation between Results and Enablers separately in three time periods. The correlation is positive if the rising in Enablers corresponds to rising in Results, or negative in opposite case.
Reviewer 2 Report
Line 3 in the Abstract:
typo : “employed effectively TO fit the incremental”
The Title –is rather ambiguous – too general, should be reformulated and more focused on the specific research.
Abstract: The abstract has a standard formulation and presents concisely the purpose, the method and the results of the research. A brief note about the methodology should be included in the abstract. There is too long description of the topic.
KEYWORDS – are accordingly.
The Introduction: tries to convince about the importance of the research scope but insists too much on descriptive issues about the Terengganu communities.
The Introduction –argues the importance of the research and states the research questions.
Theoretical framework
This section has the role of Literature review and presents the concepts in definitions and examples of practical approaches.
Description of several cases is wide but literature review of concepts should be enriched in references for supporting the same concept or idea/ or theory: EFQM Excellence model, Questionnaire method, Gompertz-function.
Methodology
This section is typically structured and presented.
Research methodology is presented in detail, and clearly enough in order to understand the research approach and implementation.
Sample and data collection is clearly presented.
The research instrument, the measurement method and scales are presented.
Data analysis is systematically approached and presented: EFQM Excellence Model, and the measurement scales of the RADAR scoring matrices.
Data analysis procedure is well and concisely described.
Results and discussion
The Results section is highly well presented on paragraphs related to each investigated component and according to each research question.
Presentation of results is very well detailed and argued in distinct sections.
Conclusions
Conclusions are formulated as short statements of the results obtained through the analysis …
In conclusion section Typo Error :
“culture and improve their performance. , but to do this with a focus on improving the 432 whole of society.”
Conclusions should include also theoretical contributions as the authors frame the topic into some new theoretical and practical coordinates.
References
References are rather centred on the methodological issues of the research topic. Should be improved with emblematic papers on knowledge management, leadership, organizational capabilities and other DOI 10.25019/MDKE/5.4.07 ; doi:10.3390/su12041348.
Author Response
Dear reviewer 2,
thank you for your valuable review. We have tried to take all your comments into account.
Reviewer 2: English language and style are fine/minor spell check required.
Response: the typos in the article have been corrected and the text has been proofread by a native English language expert.
Reviewer 2: The Title –is rather ambiguous – too general, should be reformulated and more focused on the specific research.
Response: The title has been modified to help quickly understand the focus of the article.
Reviewer 2: A brief note about the methodology should be included in the abstract. There is too long description of the topic.
Response: We have shortened the abstract based on our opponent's opinion. The methodological approach (Gompertz curve, sample method, forecasting) is already covered in the abstract and explained more in a separate section.
Reviewer 2: the research scope but insists too much on descriptive issues about the Terengganu communities.
Response: the comment is not related to the topic of the article, so we have not taken it into consideration.
Reviewer 2: literature review of concepts should be enriched in references for supporting the same concept or idea/ or theory: EFQM Excellence model, Questionnaire method, Gompertz-function.
Response: All concepts are also explained in separate subsections 2.1, 2.2 and 2.3, and are underpinned by multiple sources. We have at least extended the description and justification of the Gompertz model.
Reviewer 2: Conclusions should include also theoretical contributions as the authors frame the topic into some new theoretical and practical coordinates.
Response: We have extended the theoretical reasoning in the Conclusions based on the reviewer's recommendation.
Reviewer 2: References are rather centred on the methodological issues of the research topic. Should be improved with emblematic papers on knowledge management, leadership, organizational capabilities and other DOI 10.25019/MDKE/5.4.07 ; doi:10.3390/su12041348.
Response: doi:10.3390/su12041348 source was added and background and context of the article are enhanced. The second source DOI 10.25019/MDKE/5.4.07 has the wrong DOI.
Kind regards,
Kristina Zgodavova